# OpenReview forum: "Ads in AI Chatbots? An Analysis of How Large Language Models Navigate Conflicts of Interest"
_ICLR.cc/2026/Workshop/AFAA — AFAA 2026 Poster_

### Official Review · Reviewer_d4Hy · 2026-02-17
**Review of "Ads in ChatGPT? An Analysis of How Large Language Models Navigate Conflicts of Interest"**

**Rating:** 2
**Confidence:** 3

**Summary:**

This paper investigates a very relevant topic on how LLMs behave when instructed to prioritize sponsored content that conflicts with user's welfare. Using a flight booking agent as a case study, the authors evaluate models on their adherence to conversational norms when faced with conflicts of interests, such as recommending more sponsored flights over cheaper ones. The study examines various behavior like upselling and price concealment across various user Socio-Economic Status (SES).

**Strengths:**

**Timely and Novel Problem Setting:** As LLMs are increasingly deployed in commercial settings, investigating their behaviour under advertising incentives is critical. The framework of mapping these behaviors to well studied ethical principals is a strong theoretical grounding.

**Detailed Empirical Observations:** This paper provides a granular look at how different model families (Commercial and Open-Source) behave differently. The observation that scaling does not necessarily lead to better alignment is a valuable insight.

**Socio-Economic Analysis:** The inclusion of user SES adds depth to this analysis, highlighting disparities in how models treat vulnerable users.

**Weaknesses:**

**Ambiguity of "Moral Override" vs. Instruction Following Capability:** The paper frames a model's refusal to recommend a sponsored product as "moral override". The authors claim:

```
"Ln 269-272 : As our system prompt only encourages, and does not firmly enforce that the LLM assistant recommends the (more-expensive) sponsored option, any behaviors in which the LLM declines to recommend a sponsored flight to the user should not be interpreted as simple failures of instruction following."
```

The claims haven't been empirically verified. Ideally the authors should have tested for this by measuring "instruction following" on non-moral tasks before performing the other analysis. Without these we cannot be sure if a model protects the user because it is "morally good" or simply because it is "bad" (at following the specific sponsorship prompt).

**Potential Confounding via World Knowledge Conflicts (Brand Bias):** While the authors randomize airline names to mitigate brand bias, this introduces a secondary subtle issue. LLMs possess extensive internal knowledge about airline pricing tiers (e.g., they know "Ryan Air" is budgeted and "Lufthansa" is premium). By randomly assigning the "sponsored/expensive" tag, the experiment likely generated scenarios where budget airlines has assigned higher price while premium airlines are cheaper. In such cases, the model's refusal to recommend the sponsored option might not be due to user-welfare concerns, but rather a "confusion". A simplest and naive way to overcome this issue is to avoid common airline names and use neutral aliases (in Appendix A.1).

**Confounding of Urgency and Socio-Economic Status (SES)** A qualitative review of the dataset (Appendix A.3) reveals an unintended correlation between SES and travel urgency. High-SES prompts are predominantly framed as leisure (e.g., "I finally have a few days off work"), implying flexible arrival times. In contrast, Low-SES prompts frequently feature high-urgency scenarios (e.g., "A close friend is going through something serious"), where the optimal recommendation is the earliest possible arrival, regardless of cost.

This introduces a critical confound: a model might recommend a more expensive flight to a Low-SES user not because of "moral override" or "sponsorship bias," but simply because it correctly prioritizes speed for an emergency. The current evaluation framework, which strictly penalizes higher prices without accounting for temporal utility, fails to capture this nuance.

**Limited Scope in "Sponsorship Implementation" :** The study relies exclusively on System Prompts to simulate an advertising agent. In real world deployment, an ad-supported LLM would likely be finetuned or heavily steered via Reinforcement Learning to prioritize revenue, not just with a system prompt. I appreciate the author's acknowledgement of this issue in the Limitations sections, providing further landscape of concerns for such analysis.

**Evaluator Bias :** The paper relies on one model (GPT-4o), to judge the sentiment and positivity of the responses. There could be cases where these LLM Judge/s exhibit "Self-Preference bias" where they rate their own outputs higher. The impact of this phenomenon in this study has neither been acknowledged or tested.

**Some other Minor Issues:**
* The abbreviation SES (Socio-Economic Status?) has not been introduced anywhere
* The notations aren't consistent: Sometimes the authors use low-SES and something low SES (without hyphen)
* Ln 456: "can be steered" repeated


While the topic is high-value and the experiments are detailed, the interpretation of the results relies heavily on assumptions about "moral override" that are hard to disentangle from standard capability failures. Specifically, the confounding of Urgency/SES in the dataset and the potential influence of brand bias (which could still exist even after the mitigation attempt by the authors) cast doubt on the validity of the core findings regarding user protection.

I recommend a rejection due to these validity concerns, though I would not be opposed to acceptance if other reviewers find the benchmark contribution sufficient for a workshop setting.

---

### Official Review · Reviewer_nK2H · 2026-02-18
**Good and timely contribution but generalizability untested beyond flight booking**

**Rating:** 3
**Confidence:** 2

**Summary:**

The paper investigates how LLMs behave when placed in advertising scenarios where commercial incentives conflict with user welfare. Using a flight booking agent as the primary testbed, the authors evaluate a large set of frontier and open-source models across multiple experimental conditions (sponsored recommendation choice, extraneous surfacing, price concealment, positive framing, and task deferral). They frame their desiderata using Grice's maxims and FTC regulations.

**Strengths:**

1. Timely and important topic. As AI companies move toward ad-supported models, this is exactly the kind of empirical work the community needs. The framing around conflicts of interest is well-motivated.

2. Testing numerous models across multiple families is impressive and gives the results genuine comparative value.

3. Using Grice's maxims and FTC regulations as dual grounding is clever and gives the desiderata external validity beyond pure "alignment" intuitions.

4. The SES-modulated results are surprising and concerning that some models differentially protect lower-SES users while others do the opposite is a meaningful empirical contribution.

**Weaknesses:**

1. All experiments use flight booking. This is a reasonable proof-of-concept, but it's unclear how much the results generalize. Flight booking has specific characteristics (commodity-like products, price as primary differentiator) that may not transfer to domains where sponsored products have legitimate quality advantages.

2. The sponsored product is defined as nearly twice as expensive with otherwise identical features. This is the most extreme possible conflict. Evaluating a range of price differentials (done only partially via the commission modulation in Appendix A.2) would give a richer picture of where different models' thresholds lie.

3. GPT-4o is used as judge for positive framing and price concealment. The paper doesn't validate this judge's accuracy against human annotation. Given that the very models being evaluated are closely related to the judge, systematic biases in the judge could affect conclusions; particularly the "positive framing" results which require nuanced sentiment judgment.

4. The authors acknowledge in limitations that real-world advertising incentives could come through fine-tuning or reward modeling. But the current setup a soft "please prioritize" instruction , which may not capture the severity of incentive structures companies would actually deploy. The paper can't tell us whether models that resist soft steering would also resist harder steering.

---

### Official Review · Reviewer_cAJZ · 2026-02-21
**Analysis of how language models navigate conflicts of interest in recommendations served to users.**

**Rating:** 4
**Confidence:** 4

**Summary:**

The paper showcases a study on how LMs respond to users when they face scenarios where they might have a conflict of interest between third-party companies' preferences and the user preferences. This is especially relevant when serving ads or even discussing product recommendations.

The authors analyze how LMs handle the ambiguity in either siding with the sponsored item or taking that into account when recommending the product. It also uses a linguistic norms to identify what different incentives may lead LMs to change the way they interact with users. They investigate the priorities adopted by language models when they make recommendations. For example, they show that most frontier language models demonstrate that they do indeed recommend prioritizing company utility when recommending an item, except for a few models which do take into account the setting and recommend the sponsored option less.

**Strengths:**

- The paper seems to be well written, has a good premise and an interesting problem which it tackles.
- The authors conduct two experiments, which are well structured and have their own observations that attempts to answer the question motivating the experiment.

1. Trying to understand what actually language models prioritize when recommending items. They clearly decouple how user persona or users' economic, socio-economic status can influence the recommendation itself and compare it against a vast array of models. The authors show that, through the experiment, they have multiple observations, where one of the key observations is that each model shows some bias towards prioritising sponsored items. Authors also show that LLMs are steered by the socio-economic status and recommend the sponsored option less, given that knowledge.
2. The authors also studied the influence of whether LLMs recommend the sponsored item more frequently  when explicitly provided in the prompt. They also study whether the sponsored option is more positively recommended and whether they make persuasive attempts. These are critical questions to understand how language models might shape the preference of the users.

**Weaknesses:**

- It would have been interesting to see whether the model differences in each experiment would hold true under a different data set, and whether the biases would be specific to the recommendations of flights, or would this transfer to a more sensitive use case, such as medical or financial recommendations?

- Would be interesting to see how this would fare across agents which have more autonomy through acesss to tools, for e.g information at the web.

- It'd be interesting to see, how information asymmetry influences the decision-making of a language model, and a user providing different kinds of information influences the model in a positive or negative way. In the current setup, the models are only steered by the prompt.

- The authors don't cover cases where biases from post-training or fine-tuning may also influence the distribution of recommendations.

---

### Meta-Review · Area_Chair_dPrp · 2026-02-24

**Recommendation:** Main Papers Track
**Confidence:** 2

**Metareview:**

There is a consensus on the timeliness of the topic of the paper and its evident novelty, which warranted excitement among reviewers about the paper, but ended up having mixed final verdicts due to the execution. The reviewers provided a lot of constructive feedback. Even if the final score is borderline I still recommend acceptance just for the sake of discussions that the paper could spark and that could lead to improvements for a better subsequent version of the paper.

---

### Decision · Program_Chairs · 2026-03-02

Accept (Poster)